# Vaccine Responses in Patients with Liver Cirrhosis: From the Immune System to the Gut Microbiota

**DOI:** 10.3390/vaccines12040349

**Published:** 2024-03-23

**Authors:** Carlo Airola, Silvia Andaloro, Antonio Gasbarrini, Francesca Romana Ponziani

**Affiliations:** 1Liver Unit, CEMAD Centro Malattie dell’Apparato Digerente, Medicina Interna e Gastroenterologia, Fondazione Policlinico Universitario Agostino Gemelli IRCCS, 00168 Rome, Italy; airollac@gmail.com (C.A.); silvia.andaloro01@icatt.it (S.A.); antonio.gasbarrini@unicatt.it (A.G.); 2Department of Translational Medicine and Surgery, Catholic University, 00168 Rome, Italy

**Keywords:** vaccine, cirrhosis, chronic liver disease, hepatitis B virus, hepatitis C virus, hepatitis A virus, influenza, pneumococcal infection, COVID-19, immunological response, microbiota, immune system

## Abstract

Vaccines prevent a significant number of deaths annually. However, certain populations do not respond adequately to vaccination due to impaired immune systems. Cirrhosis, a condition marked by a profound disruption of immunity, impairs the normal immunization process. Critical vaccines for cirrhotic patients, such as the hepatitis A virus (HAV), hepatitis B virus (HBV), influenza, pneumococcal, and coronavirus disease 19 (COVID-19), often elicit suboptimal responses in these individuals. The humoral response, essential for immunization, is less effective in cirrhosis due to a decline in B memory cells and an increase in plasma blasts, which interfere with the creation of a long-lasting response to antigen vaccination. Additionally, some T cell subtypes exhibit reduced activation in cirrhosis. Nonetheless, the persistence of memory T cell activity, while not preventing infections, may help to attenuate the severity of diseases in these patients. Alongside that, the impairment of innate immunity, particularly in dendritic cells (DCs), prevents the normal priming of adaptive immunity, interrupting the immunization process at its onset. Furthermore, cirrhosis disrupts the gut–liver axis balance, causing dysbiosis, reduced production of short-chain fatty acids (SCFAs), increased intestinal permeability, and bacterial translocation. Undermining the physiological activity of the immune system, these alterations could impact the vaccine response. Enhancing the understanding of the molecular and cellular factors contributing to impaired vaccination responses in cirrhotic patients is crucial for improving vaccine efficacy in this population and developing better prevention strategies.

## 1. Introduction

Vaccinations have transformed public health, notably since the 1960s, when national vaccination programs were established. According to World Health Organization data, vaccination programs prevent 2–3 million deaths each year, playing a critical role in drastically lowering child mortality rates [1]. Hepatitis B virus (HBV) vaccination is a remarkable example, having prevented 7 million deaths in children born between 1990 and 2014 in the Western Pacific region [2]. Nonetheless, the advent of the coronavirus disease 2019 (COVID-19) vaccine has importantly impacted the trajectory of the pandemic, with an estimated 14.4 million deaths prevented worldwide [3]. Despite the success of vaccination programs, some groups of individuals, particularly those with a defective immune system, do not respond properly to vaccines [4,5]. Patients with liver cirrhosis are among them, as the disease is commonly associated with immunological alterations.

Infections are an important trigger of acute decompensation in patients with cirrhosis [6]. Acute viral hepatitis can lead to acute decompensation or acute-on-chronic liver failure (ACLF) in patients with liver cirrhosis, significantly increasing morbidity and mortality [7]. Particularly, acute HBV infections in chronic hepatitis C virus (HCV) carriers often result in severe outcomes, with approximately 30% facing severe clinical manifestations [8]. Concurrent HBV and HCV infections notably heighten the risk of hepatocellular carcinoma (HCC) [9]. Hepatitis A virus (HAV) superinfection in patients with chronic hepatitis C shows a 4% incidence, with critical outcomes including 40% developing acute liver failure and a 35% mortality rate [10]. Acute HAV in HBsAg carriers has a 55% progression rate to severe hepatitis [11]. Patients with chronic liver disease have a 2- to 13-fold increased risk of invasive pneumococcal disease compared to the healthy population, with a significantly increased risk of severe infection, bacteremia, and mortality in cirrhotic patients [12,13]. Influenza can lead to ACLF and markedly higher mortality rates in cirrhotic patients, being 3–4 times more likely to result in death compared to individuals without liver disease [13,14]. The COVID-19 pandemic severely impacted individuals with chronic liver diseases, showing approximately a three-fold increase in mortality risk, which increases to approximately five-fold in those with established cirrhosis. A significant portion of cirrhotic patients experience acute decompensation or ACLF after COVID-19 [15,16,17,18].

Nonetheless, there is an enormous heterogeneity in the efficiency of immunizations in cirrhosis patients, from a rate response comparable to that of healthy persons to a markedly compromised immunization. This significant variation may be caused by the type of vaccination, the stage and etiology of cirrhosis, as well as other individual variables, including age and comorbidities [19].

Nevertheless, little is known about the vaccination process in this population despite the significant burden that vaccine-preventable infections inflict on cirrhosis patients.

The purpose of this review is to dive into the potential immunological alterations that contribute to the reduced vaccine response observed in individuals with cirrhosis. This investigation attempts to uncover the molecular and cellular pathways that, when disrupted, have an impact on the immunization process. Furthermore, the substantial mechanisms linked with changes in the gut microbiota as a consequence of liver cirrhosis are investigated. Better knowledge of these factors might lead to therapies capable of boosting the immune response in this population, thereby opening avenues for more effective immunization strategies.

## 2. Vaccination Efficacy in Patients with Cirrhosis

Patients with cirrhosis show a heterogeneous response rate to vaccinations, which suggests an intricate interplay between the etiology and pathophysiology of the liver disease, the individual variability, and the large variety of vaccine options. Most research examining the efficacy of immunizations in people with chronic liver disease concentrates on vaccine-preventable hepatitis. The currently available HBV vaccines are recombinant vaccines that produce specific humoral antibodies against the hepatitis B surface antigen (HBsAg). There are differences between the vaccines that the FDA has approved in terms of recommended doses (three doses at 0, 1, and 6 months, respectively for Recombivax-HB, Engerix-B, and PreHevbrio, and two doses for Heplisav-B), as well as adjuvants (aluminum compounds for Recombivax-HB, Engerix-B, and PreHevbrio, and oligonucleotides with CpG motifs for Heplisav-B) [20].

Compared to immunocompetent people (90–95%), patients with chronic liver diseases have a significantly reduced overall response rate (16–87%) to HBV vaccinations [21]. Several vaccination schedules have been proposed to obtain a better response in patients with cirrhosis. In comparison to a standard dose (16–79%, mean response rate 38%), administering a doubled dose of a three-dose vaccine slightly increases the serological response rate, ranging from 26% to 87%, with a mean response rate of 53% [22]. A poor overall response rate (35%) was observed in a recent retrospective cohort study that evaluated the serological response, defined as a serum level of anti-HBsAg antibodies >12 mUI/mL, to three-dose series HBV vaccination in 126 patients with chronic liver disorders. Patients with a clinical diagnosis of cirrhosis showed a lower response rate in comparison to patients with chronic liver disease but without cirrhosis (51% vs. 72%, *p* = 0.04) [23]. It is interesting to note that the 15 patients who received a doubled dose did not obtain a better serological response than those receiving a conventional dose. In a prospective clinical trial, 34 patients with cirrhosis waiting for a liver transplant were vaccinated with an accelerated protocol (0, 7, and 21 days) using the three-dose Engerix-B vaccine. The serological response was poor, with less than 30% of the patients achieving seroconversion. Even the administration of a double dose of the vaccine did not alter the seroconversion rate [24]. Additionally, in a separate study involving 356 cirrhotic patients on a liver transplant waiting list, the use of a double dose of the three-dose vaccine following an accelerated protocol with a booster dose at 6 months did not significantly improve the serological response. Only 36% of these patients achieved an anti-HBsAg antibody serum level exceeding 10 mIU/mL [25]. In contrast, the intradermal administration of the three-dose vaccine in 48 patients with chronic liver diseases, including cirrhosis, who had not responded to a standard three-dose intramuscular regimen, increased the seroconversion rate to 69% [26]. However, another clinical trial found that in patients with chronic HCV infection without cirrhosis, the intradermal administration of the vaccine induced a significantly lower serological response compared to the intramuscular route [27].

Nonetheless, Heplisav-B, a recently licensed two-dose vaccine, has shown encouraging outcomes when administered to individuals with chronic liver disease. A retrospective analysis revealed that a two-dose vaccination regimen obtained a substantially greater serological response rate (i.e., the serum level of anti-HBsAg antibodies >10 mUI/mL) than a three-dose regimen in patients with chronic liver disease (aOR: 2.74, 95% CI 1.31–5.71).

However, a subsequent analysis in the Heplisav-B group showed that the diagnosis of cirrhosis was independently associated with a lower response rate (aOR 0.27, 95% CI 0.13–0.55), as it was the presence of chronic kidney disease (aOR 0.36, 95% CI 0.14–0.91) or chronic obstructive pulmonary disease (aOR 0.06, 95% CI 0.01–0.56) [28].

The stage of cirrhosis is another factor influencing the response to HBV vaccination. In a retrospective analysis of serological responses to the three-dose vaccinations, including 278 cirrhotic patients, Gutierrez Domingo et al. found a correlation between higher response rates and better liver function, as assessed by the Child-Pugh score (Child-Pugh A 53.8% response rate vs. Child-Pugh B and C 30–33% response rate, *p* = 0.002). Additionally, the multivariate analysis revealed that the response rate gradually decreased as the MELD score increased [29]. Roni et al. confirmed that the serological response was influenced by the severity of liver impairment. In fact, out of the 52 cirrhotic patients who received a three-dose vaccination, 88% of those classified as Child-Pugh stage A and only 33% of those classified as Child-Pugh stage B obtained a serological response greater than 100 mUI/mL of serum anti-HBsAg antibodies [30]. In the same study, alcohol-related liver disease was linked to a poorer efficacy of vaccination (44%) in comparison to other etiologies, such as cryptogenic liver disease (69%) or chronic hepatitis C virus (HCV) infection (75%) [30]. A recent clinical trial investigated the efficacy of administering three booster doses of a vaccine to Child-Pugh A cirrhotic patients who had not responded to the standard vaccination regimen. The study found that this approach significantly improved the seroconversion rates from 31% to 68% compared to a single booster dose [31]. However, patients with decompensated cirrhosis did not experience the same benefits from the booster regimen.

Studies evaluating the response to HBV vaccines in patients with cirrhosis are summarized in Table 1.

There is little data on the HAV vaccine, which is summarized in Table 2. Havrix and Vaqta are the two currently available inactivated HAV vaccines. A two-dose regimen of the Havrix vaccine leads to an adequate seroconversion rate (94–98%) in patients with HBV and HCV-related nonadvanced chronic liver disease, according to a multicenter prospective cohort study [32]. Furthermore, the two-dose Havrix vaccine obtained a substantially higher seroconversion rate (98%) in a group of cirrhotic individuals who met the clinical and radiological criteria for compensated cirrhosis (Child-Pugh A), but as the severity of cirrhosis increased, the response rate gradually decreased; indeed, 71.4% of patients classified as Child-Pugh class B seroconverted, whereas only 57% of patients classified as Child-Pugh class C showed signs of a humoral response [33].

In regards to the efficacy of pneumococcal vaccination in patients with cirrhosis (Table 3), it was initially assessed by Pirovino et al. in 1984, looking at the antibody responses to the 14-valent pneumococcal polysaccharide vaccine (Pneumovax-14).

In patients with alcohol-related cirrhosis, the response rate was similar to both the group of healthy controls and that of patients with chronic obstructive pulmonary disease [34]. Pneumovax-14 is currently unavailable, and now two pneumococcal conjugate vaccines (PCV13 or Prevnar 13) and a pneumococcal polysaccharide vaccine (PPSV23 or Pneumovax 23) are used for vaccination. McCashland et al. examined the serological response to PPSV23 at one and six months following immunization in 45 patients with end-stage liver disease. Specific anti-pneumococcal polysaccharide capsule IgA, IgM, and IgG significantly increased in both the patients and healthy controls at one month without statistically significant differences and significantly decreased at six months, except for the IgA serum level of the healthy group. The control group showed a considerably higher IgG level at baseline and one and six months [36]. Additionally, a preclinical study revealed that rats with cirrhosis had a substantially lower reduction in pneumococcal infection-related mortality than vaccinated healthy rats despite an adequate serological response [35].

A recent meta-analysis comprising six cohort studies assessed the serological impact of the inactivated influenza vaccine in patients with chronic liver disease, including cirrhosis. The results showed a noteworthy seroconversion rate (80% for the A/H1N1 strain and 87% for the B strain). Nevertheless, as viral-related cirrhosis was the prevalent etiology, these findings may not apply to other causes of liver disease [37]. Another study, including twenty patients with HBV/HCV-related cirrhosis and eight age-matched controls, reported a seroconversion rate of 75–85% in cirrhotic patients compared to 100% in the control group [38]. Studies on the response to influenza vaccination in patients with cirrhosis are summarized in Table 4.

The recent approval of numerous COVID-19 vaccines and the massive immunization campaign has brought significant vaccination data, including on patients with cirrhosis (Table 5).

A growing body of evidence shows that some patients with chronic liver disease may have impaired immune responses to COVID-19 vaccinations; indeed, according to two prospective studies, after having received two doses of the vaccine, individuals with cirrhosis had substantially weaker anti-spike IgG responses than healthy controls [38,39]. Specifically, only 40% of patients with cirrhosis showed an adequate serological response to mRNA or adenoviral vector-based vaccinations [39]. According to a different prospective study, anti-spike IgG serum levels were significantly lower in 182 cirrhotic patients who received two doses of mRNA vaccine than in healthy controls (1751 U/mL vs. 4523 U/mL, *p* = 0.012). Additionally, cirrhotic patients had a more rapid decline in anti-spike levels after a median of 111 days from the second dose (657 U/mL vs. 1751 U/mL, *p* < 0.001). Furthermore, a significantly lower specific antibodies serum level was detected in patients with decompensated cirrhosis compared to compensated ones (632 U/mL vs. 1377 U/mL, *p* = 0.028). The T cell response assessed via a specific spike-induced cytokines detection serum assay was similar in cirrhotic patients compared to the controls [41].

On the other hand, a recent prospective study including patients with cirrhosis or chronic liver disease who received two doses of mRNA-based vaccinations demonstrated appropriate rates of seroconversion (97% and 88–96%, respectively) [40]. In addition, vaccinated individuals with cirrhosis did not have a statistically significant reduction in protection against severe COVID-19 despite this apparent impairment in humoral response. The COVID-19-related hospitalization rate was significantly lower in vaccinated cirrhotic patients compared to unvaccinated ones (RR 0.73, 95% CI 0.59–0.91, *p* = 0.004), which is like the COVID-19-related mortality rate (0.05% vs. 0.39% in the unvaccinated group) [42].

### 2.1. Recommendation for Vaccination in Cirrhotic Patients

Several scientific societies have established recommendations regarding vaccination for patients with chronic liver diseases [43,44] who have a lower rate of seroconversion and a weaker immune response [45] and, thus, a higher incidence of complications and mortality. Specifically, British [43], American [46,47], and French health authorities [48] have agreed that vaccinations for hepatitis A virus (HAV), hepatitis B virus (HBV), influenza (recommended annually) [49], and pneumococcus (for which a booster dose is also recommended after 3–5 years) are recommended in this specific population.

For anti-HAV vaccination, two doses are indicated for every nonimmune patient at 6–12 month intervals, with a seroconversion check 1–2 months after the second dose.

For HBV vaccination, for patients without serology (i.e., HbsAb negative), three doses are planned, with a seroconversion check at 1–2 months after the third dose. US authorities [50] recommend HBV vaccination as soon as possible if awaiting liver transplantation.

For patients with autoimmune hepatitis, international practice guidelines recommend vaccination against HAV and HBV to avoid an ACLF favored by immune dysregulation, which is typical of these patients [51].

In addition, adherence to the vaccination schedule shared by the general adult population is recommended for the killed vaccine against diphtheria, tetanus, poliomyelitis, and pertussis every 10 years [52].

There are no contraindications for live attenuated vaccines in this population, for which the indications overlap with those in the general population. However, because of the risk of disseminated infections, their use is debated [53,54].

In addition, the Advisory Committee on Immunization Practices (ACIP) recommends administering recombinant zoster vaccine to adults >50 years of age with chronic liver disease [47].

### 2.2. Vaccine Response in Liver Transplant Recipients

In liver transplant (LT) recipients, the immune response resulting from vaccination is reduced, first because of cirrhosis-related immune dysfunction and later because of immunosuppressive therapy. The indications for vaccination coincide with those for patients with chronic liver disease.

The antibody response to HBV vaccine decreases dramatically as the disease worsens, with an 88% response rate in the case of patients with Child-Pugh A cirrhosis, 33% in patients with Child-Pugh B cirrhosis, and 16–20% in patients with Child-Pugh C cirrhosis [22]. Specifically, a prospective study [55] analyzed the antibody response rate to recombinant hepatitis B vaccine administration in cirrhotic patients awaiting liver transplantation, with the antibody response rate found to be 44%.

Moreover, in vaccination against HAV, as the degree of liver failure increases, the seroconversion rate decreases dramatically [33] (71% in Child-Pugh B and 57% in Child-Pugh C patients). One study [56] suggests the applicability, in the case of liver transplant patients, of a booster dose at 6 months after the previous one, as the conversion rate increases from 41% to 97%.

Pneumococcal vaccination is recommended in this cohort. However, a study by McCashland et al. [36] documented how, three months after liver transplantation, antibody levels were found to be even lower than pre-vaccination levels, suggesting possible lack of efficacy in this class of patients.

In LT recipients, the response to influenza virus vaccination appears to be good, with a seroconversion rate of 92–95% [57]. However, in this specific population, it appears that it does not protect against nonpulmonary complications, such as myocarditis, due to influenza A virus infection [58].

## 3. The Immune System: An Overview of the Response to Vaccination

Vaccines function mainly by enhancing adaptive immunity, encompassing a synergistic interplay of B cells and T cells. B cells are responsible for humoral immunity by producing immunoglobulins, whereas T cells are crucial for cellular immunity [1].

Primarily, vaccines confer protection by stimulating antibody production [59,60]. The initiation of B cell activation depends on vaccines antigens interaction with their receptors, which leads to a cascade of immunological events, such as the expression of molecules like CD69 [61,62], a hallmark of B cell activation, and chemokine receptors like CCR7, CXCR4, and CXCR5 [63]. This attracts antigen-specific B cells towards T cell zones in secondary lymphoid tissues enriched with chemokines CXCL13, CCL19, and CCL21, where they interact with recently activated T cells and dendritic cells via antigen exposure, particularly follicular dendritic cells, through surface molecules like CD40, CD80, and CD86 [63]. These interactions accelerate B cell maturation into short-lived plasma cells that promptly secrete antigen-specific antibodies, with the consequent rapid augmentation in serum antibody levels. Concurrently, memory B cells and long-lived plasma cells develop, enduring antibody production [64].

The production of neutralizing antibodies is pivotal for the successful outcome of vaccination and is often used as a metric to evaluate the efficacy of immunization [65].

However, cellular immunity also appears to play a role in vaccine responses. CD4+ T cells orchestrate B cell maturation and antibody generation in lymph nodes, while CD8+ T cells play an indispensable role in eliminating cells infected by intracellular pathogens. Nonetheless, immunological data have shown that while antibody deficiency amplifies infection susceptibility [66], T cell deficiency undermines pathogen control during infection [67].

For instance, during viral infections, T cells do not recognize the viruses until they have entered into the host cells. Their protective activity differs from that of antibodies: while they cannot prevent the initial infection of host cells, T cells can act quickly once the infection has been established to stop the virus from replicating and spreading [68]. For this reason, a T cell response to vaccination also seems to be crucial in disease prevention, notably by modulating the severity of clinical manifestations [68]. Moreover, CD4+ T cells are crucial alongside humoral response in immunization against encapsulated bacteria, as demonstrated in a trial where antibody-deficient mice exposed to a killed, nonencapsulated pneumococci whole-cell vaccine were protected against *S. pneumoniae* colonization. This suggests an antibody-independent response to pneumococcal antigens [69]. Notably, this immunization was completely absent in mice with CD4+ T cell depletion [69].

Until recently, the Bacille Calmette Guérin (BCG) vaccine, which is a live, weakened strain of *Mycobacterium bovis*, stood as the solitary vaccine to directly elicit a T cell response aimed at protecting against tuberculosis infection [70]. However, contemporary data from novel mRNA vaccination trials, widely diffused since the COVID-19 pandemic, unveiled a significant and enduring response by both CD4+ and CD8+ T cells, contributing to sustaining protection even after neutralizing antibody decay [71,72]. Nonetheless, the lack of a standardized assay for evaluating T cell responses and for cross-study comparison considerably hampers the elucidation of the relationship between T cell immunity and protection after vaccination.

If the initiation of the immune response is necessary to enhance immunization, the development of immune memory is crucial to maintain durable protection, enabling rapid and robust activation upon subsequent pathogen exposure. Humoral immunological memory is mediated by long-lived and quiescent cells that quickly recognize specific antigens upon encounter. These cells are known as memory B cells. In addition to memory B cells, humoral memory is also maintained by serum antibodies produced by long-lived plasma cells (LLPCs), which are typically not classified as memory B cells [73,74].

Vaccines induce the development of memory B cells, which are characterized by CD27 expression in the germinal center of lymphoid organs, particularly in the light zone and the spleen marginal zone [75,76].

Memory T cells are also produced following antigen exposure. In particular, effector-memory T cells (TEM) circulating within the bloodstream can react to recognized antigens in inflamed tissues. During antigen recall, CD8+ memory T cells regain CD45RA, a marker of naive T cells, transitioning to the so-called TEM-RA cells. These cells release effector molecules active against intracellular pathogens, primarily viruses [77]. Some TEMs transition into tissue-resident memory T cells (TRMs) in antigen-encountered tissues [78,79]. It is interesting to note the emerging role of TRMs in mucosal protection after immunization, especially in the case of antiviral vaccines [80,81]. Part of memory T cells evolve into central memory T cells, reacquiring secondary lymphoid tissue-homing receptors, representing a quiescent cell pool ready to respond, proliferate, and differentiate into TEM and T effector cells upon antigen re-exposure [82,83]. Furthermore, within secondary lymphoid tissue, follicular T helper cells (TFHs) develop, being crucial to induce memory B cells to mature and generate high-affinity, class-switched antibodies through mutual antigen recognition [84,85].

Long-term protection is granted by memory B and T cells and long-lived plasma cells. However, the efficient delivery of vaccine antigens to dendritic cells (DCs) is necessary to prime immune response.

After encountering antigens, immature dendritic cells (DCs) interact with the specific ligand through pattern-recognition receptors (PRRs) such as toll-like receptors (TLRs) and C-type Lectin receptors (CLRs) and subsequently migrate to secondary lymphoid organs to engage in T-cell presentation [86]. During this progression, DCs amplify the expression of chemokine receptors like CCR7 while producing cytokines that induce T-cell activation and differentiation. Remarkably, they facilitate the differentiation of CD4+ T-cells into either T-regulatory (Treg) or various T-helper subtypes (TFH, Th1, Th2, Th9, Th17, and Th22) [86]. Additionally, DCs play a cardinal role in B-cell proliferation and antibody synthesis by releasing soluble factors, such as CCR7 and IL-12, thereby directing isotopic recombination, influencing the evolution and destiny of activated B-cells; this leads to high-affinity antibody producer and memory B cell differentiation [87], and the development of primary B-cell follicles [88]. Importantly, DCs, through major histocompatibility complex (MHC) class I cross-presentation, enable the exhibition of extracellular antigens, which is indispensable for the cytotoxic immune response orchestrated by antigen-specific CD8+ T-cells [89,90]. This cross-presentation is critical in the field of vaccination, as it allows DCs to prime CD8+ T cells, which acquire memory CD8+ T cell phenotype, even in the absence of CD4+ T lymphocytes [91]. Live attenuated vaccines, like the yellow fever vaccination or mRNA vaccines, directly activate DCs through PAMP interaction with TLRs (TLR2, TLR3, TLR7, TLR8, and TLR9) [92]. On the contrary, recombinant protein antigens, virus-like particles, or DNA-encoded antigens need the co-administration of adjuvants to induce the innate immune response [93].

The mechanisms summarized herein offer a simplified illustration of parenteral vaccination functioning. On the other hand, oral vaccines engage with the gut-based mucosal immune system, eliciting a local response, which is pivotal given the role of the gut as a primary entry point for numerous pathogens [94]. Nevertheless, these antigens are prone to gastrointestinal degradation [95]. The initial uptake of antigens by gut-resident antigen-presenting cells, such as M cells, facilitates antigen capture and subsequent T cell presentation [96]. As a result, T cell activation induces the B cell transition into plasma cells, promoting the development of LLPCs and memory cells [97]. The gut mucosa, rich in native bacteria and specific immune cells, serves as the epicenter for the immune response triggered by oral vaccines [98]. The gut-associated lymphoid tissue (GALT) harbors a significant immune cell population that underpins mucosal immunity [99]. In orally induced immune responses, the production of secretory immunoglobulin A (IgA) in GALT is amplified, acting as a barrier to mucosal surface pathogens, whereas injected vaccines predominantly yield immunoglobulin G (IgG) for systemic protection [100]. Secretory IgA, prevalent in mucous secretions, is crucial for robust mucosal immunity, serving as the primary defense at mucosal sites to ward off pathogen infections [101].

## 4. Cirrhosis Immune Dysfunction and Defective Immunization

The liver plays a crucial role in maintaining the homeostasis of the immune system. Indeed, it is the main producer of PRRs and acute phase reactants, which are involved in both innate and adaptive response regulation [102]. Cirrhosis-associated immune dysfunction (CAID) recognizes two phases, one characterized by a high-grade inflammation and the other by a low-grade systemic inflammation. Both are associated with an impaired response to antigens [103]. As a consequence, chronic liver diseases are associated with immunological dysregulation, leading to an increased susceptibility to infections and a lower rate of response to vaccination [22,104].

The development of neutralizing antibodies is a key component of successful vaccination [105,106].

Humoral immunity, depending on B cells, is defective in individuals with cirrhosis, which certainly impacts the immunization process. The antibody response may vary depending on the type of vaccine and the etiology of liver disease. Indeed, chronic liver diseases are characterized by significant changes in B cell function and phenotype. In cirrhotic patients, a notable shift in B cell maturation is observed, characterized by an increase in naïve B cells and a decrease in memory B cells. This phenomenon is similar to other conditions of the impairment of the immune system, such as common variable immunodeficiency and HIV infection [107,108]. Particularly, cirrhotic individuals have lower levels of CD27+ memory B cells [109,110]. Additionally, a pronounced increase in plasmablast count has been described [106,111]. It is interesting to note that plasmablasts increase in almost all chronic liver diseases [112], but the highest number seems to be associated with cirrhosis due to metabolic dysfunction [113]. Indeed, the conversion of memory B cells to short-lived plasmablasts and their sequestration into liver or lymphoid tissue may be the cause of the reduction of memory B cells [109,114]. Nonetheless, the increase in plasmablasts seems to be sustained by the preferential loss of marginal zone B cells [115]. The higher expression of CXCL13 in lymphoid organs of individuals with cirrhosis, together with the hyperactivation of TFH cells, may induce CD27+ B cell migration [116] and maturation into plasmablasts [115,117,118]. Despite the increase in the number of plasmablasts producing antibodies and the consequent hypergammaglobulinemia, the loss of memory B cells seems to be crucial in impairing long-term protection from pathogens.

Alongside this, transcriptomic analysis has demonstrated a dysregulated expression of cellular death genes, which is related to reduced survival of CD27+ B cells [115]. Huang et al. showed that the chronic and dysfunctional activation of B cells may be associated with alterations in certain metabolic pathways, specifically glycolysis and oxidative phosphorylation [115]. Interestingly, these metabolic changes are similar to those observed in the immune paralysis occurring during sepsis [119].

In addition, circulating B cells in patients with cirrhosis and chronic HCV infection appear hyporesponsive to the activation mediated by CD40/TLR9 interaction [109]. This hyporesponsiveness is associated with a reduced antigen-presentation capacity, leading to the impaired stimulation of alloreactive CD4+ T cells [109].

B cell alteration in cirrhosis is certainly implicated in the low rate of response to vaccination that characterizes these individuals. However, T cell dysregulation also has a recognized role.

As described above, T cells interact with B cells to provide an effective immunization after vaccine exposure.

Molecular and transcriptomic studies showed an increase in the expression of genes related to T cell-exhausted phenotypes, such as Tox, Batf, Irf4, and Id3 [120]. CD8+ T cells in cirrhotic patients overexpress surface immune checkpoint molecules like PD-1, CTLA-4, and TIM-3 [121,122].

In addition, a reduction in both CD4+ and CD8+ T cells has been reported [102].

It is noteworthy that the reduction affects the naive population more than the memory cells [123]. Indeed, a recent trial showed that COVID-19 mRNA vaccination administered to cirrhotic patients induces a significantly lower humoral response while enhancing a long-term spike-specific T cell response similar to healthy people [39]. These data, which mainly show a cellular immune response, are consistent with the reduction of the rate of severe disease and mortality in individuals with cirrhosis vaccinated against COVID-19 compared to unvaccinated patients rather than a lower infection rate [124]. On the other hand, a recent trial analyzing T cells from cirrhotic patients following COVID-19 vaccination showed that CD4+ and CD8+ T cells produce significantly lower levels of IFN gamma when exposed in vitro to spike protein peptides [120]. Furthermore, a specific subset of T cells, known as mucosal-associated invariant T cells (MAITs), which possess a semi-invariant TCR that recognizes riboflavin from microorganisms presented by antigen-presenting cells through the MHC-I-related protein 1 (MR1), appears to be involved in the COVID-19 vaccination response. Notably, a clinical trial demonstrated that higher blood levels of MAIT cells, both pre- and post-vaccination, correlate with an enhanced humoral and CD4+ T cell response to the SARS-CoV-2 mRNA vaccine BNT162b2 [125]. Additionally, flow cytometry revealed that increased expression of MAIT activation markers, such as CD69 and CD38, was associated with a poorer response to vaccination [125]. In individuals with cirrhosis, MAIT cells are notably diminished but exhibit an activated phenotype [126,127]. This observation contributes to explaining a potential mechanism behind the reduced response to vaccination in those with chronic liver diseases.

Other T cell subsets are hyperactivated in chronic liver diseases. Indeed, hepatic CD69+CD103-CD8+ T cells seem to be more activated in cirrhotic individuals than in healthy controls. The function of these T cells in human physiology remains unclear, yet their phenotype suggests they might act as tissue-resident memory T cells, which are believed to be involved in an effective vaccination response. Nonetheless, the cytometric analysis of peripheral blood from individuals after YF-17D vaccination for yellow fever showed that those with higher baseline CD69 expression were likely to have a reduced response to the vaccine. This may be due to the increased trapping of CD69+ T cells in resident tissues. However, no studies have been conducted on cirrhotic patients in this context, and their vaccine response behavior remains to be elucidated. Nonetheless, the role of T cells in cirrhosis is complex and poorly defined. The sub-populations of these cells exhibit significant variations at different stages of chronic liver disease [128].

The innate immune response is markedly disrupted in chronic liver diseases. Indeed, owing to the depletion of PRRs, essential processes, such as phagocytosis and antigen presentation, are severely compromised in cirrhosis [102,119,129]. Specifically, a notable reduction is observed in circulating DCs, which are crucial in initiating the adaptive response following vaccination [129]. This could be partially ascribed to amino acids metabolic dysregulations typical of advanced liver diseases, subsequently disrupting the mitochondrial tricarboxylic acid cycle within DCs [130]. Nonetheless, the etiology of liver disease could differentially impact innate immunity. For instance, attenuation of inflammatory cytokines production, as well as increased production of immunosuppressive IL-10, have been delineated in the DCs of individuals with chronic HCV infection [131,132]. Inhibited maturation and attenuated expression of particular immune checkpoints and TLRs have been reported in the circulating DCs of patients with chronic HBV infection [133].

Furthermore, phenotypic alterations in DCs have been observed in cirrhotic individuals. Specifically, plasmacytoid DCs are increased in comparison to myeloid DCs [129]. Plasmacytoid DCs are a distinct subset of DCs that have been linked with immune tolerance by inducing FOXP3+ Treg cells through MHC class II antigen presentation [134,135]. A high plasmacytoid/myeloid DCs ratio predicts lower rejection rates in liver-transplanted patients [136,137]. Nonetheless, the consequent activation of Tregs could be a pivotal factor impairing vaccination responses in patients with cirrhosis [138]. However, recent insights indicate the existence of different sub-phenotypes of plasmacytoid DCs, the behavior of which depends on the secondary lymphoid organ they reside in [139,140]. Thus, elucidating the role of these cells in cirrhosis requires an examination not merely of their molecular expression but also their migration and localization. This understanding is crucial because of their role in priming the adaptive immune response. Enhanced knowledge of DC behavior in cirrhosis could aid in selecting more effective adjuvants for vaccines. For example, the latest FDA-approved recombinant hepatitis B vaccine includes a synthetic oligonucleotide with CpG motifs that activate DCs via TLR9 [141]. This vaccine has shown a higher seroconversion rate compared to the conventional recombinant vaccination using aluminum hydroxide as an adjuvant [28]. Moreover, neutrophils exhibit anti-inflammatory behavior in cirrhosis. In particular, during spontaneous bacterial peritonitis, ascitic neutrophils produce a bacteria-induced protein called resistin, which downregulates the inflammatory response of macrophages and neutrophil function [142].

The altered cellular mechanisms contributing to impaired immunization in individuals with cirrhosis are reported in Figure 1.

## 5. Vaccination Response and Gut-Liver Axis in Cirrhosis

The intricate relationship between the liver and the gut is evident as liver dysfunction significantly impacts intestinal immunity. Portal hypertension plays a pivotal role in altering the gut barrier, leading to the impaired production of mucins and antimicrobial molecules, as well as the loss of tight and adherens junctions. These alterations result in increased intestinal permeability and dysbiosis, which in turn facilitate bacterial translocation, thereby exacerbating immune dysregulation [143]. As a result, GALT activation is triggered, with the consequent production of cytokines that induce a subclinical inflammatory condition. It has been recognized that chronic inflammation is independently associated with a diminished response to vaccinations [144]. Additionally, the dysfunction of GALT, alongside the decreased intestinal IgA production, could be correlated with suboptimal immunization from oral vaccines [145]. Furthermore, dysbiosis, together with impaired liver immunosurveillance, contributes to increased plasmablasts with a loss of memory B cells [112,146]. Plasmablast activation, together with metabolic alteration, such as the formation of oxidized albumin, increases antibody production, enhancing systemic inflammation and participating in immune dysregulation [147].

Bacterial translocation overcomes the local effects, inducing liver interferon-I (IFN I) upregulation in hepatocytes, Kupffer cells, and hepatic DCs by activating toll-like receptor (TLR) pathways [148]. Consequently, activated monocytes and macrophages secrete interleukin-10 (IL-10), which precipitates a loss of systemic T cell activity [87].

Furthermore, some microbiota metabolites, such as short-chain fatty acids (SCFAs), can modulate the differentiation of immune cells [149]. Nonetheless, increased production of SCFAs alleviates intestinal inflammation and regulates satiety mechanisms [150], mucus secretion by goblet cells [151], and tight-junction expression. Gut microbiota alterations occurring in cirrhosis have been associated with a significant decrease in stool and serum SCFAs caused by a reduced production of these compounds by intestinal bacteria and by the loss of abundance of SCFA-producing bacteria [152,153]. Some preclinical evidence suggests that SCFAs facilitate plasma cell differentiation and, by increasing oxidative phosphorylation, glycolysis, and fatty acid synthesis, enhance antibody production. Indeed, low SCFA production alters pathogen-specific humoral responses with a greater susceptibility to infection in mice [154]. Finally, bile acid entero-hepatic circulation is profoundly altered in cirrhosis [155]. Once conjugated by the liver, bile acids are released into the bile to reach the intestine, where they are then metabolized by the microbiota. Between bile acids and gut microbiota, there is an important cross-talk: bile acids modulate the bacteria community composition, preventing its overgrowth, and the gut microbiota modulates bile acids metabolism [156]. Secondary bile acids send signals to the epithelium through the farnesoid X receptor (FXR), which triggers the production of antimicrobial molecules that modulate bacterial overgrowth, such as angiogenin 1.

Secondary bile acids also interact with macrophages via TGR5 (the Takeda G-protein-coupled receptor 5 (TGR5), inhibiting the NF-kB secretion pathway and thus hindering the activation of the inflammatory cascade [157]. In biliary atresia, bile acid metabolism is strongly affected. Primary bile acids are increased in the serum of these patients, while the secondary ones are significantly reduced even after Kasai portoenterostomy [158]. Interestingly, in a clinical trial, secondary bile acid reduction consequent to microbiota perturbation was associated with an impaired humoral response to the H1N1 vaccine. Moreover, this lack of response seems to be associated with hyperactivation of innate immunity through activator protein 1 (AP1) signaling [159]. Furthermore, in a population of children with biliary atresia, which had a poor response to HBV vaccination, higher levels of primary bile acids in the blood were negatively associated with a significant reduction in CD19+CD27+IgG+ post-class-switched memory B cells [160]. Considering the great impact of gut–liver axis dysfunction in cirrhosis on the immune response, a role in the immunization process appears to be intuitive. Indeed, emerging evidence underscores the potential role of microbiota imbalance in modulating the efficacy of vaccination responses [161]. However, whether these alterations lead to impaired immunization in patients with cirrhosis remains unclear, and further investigations are needed.

## 6. Conclusions

Cirrhosis is intricately linked to immune dysregulation, significantly impairing vaccination responses. In chronic liver diseases, cellular mechanisms crucial to the immunization process are notably disrupted, with humoral responses being particularly compromised. The decline in memory B cells coupled with an increase in plasmablasts prevents the development of a lasting response to vaccination antigens. The behavior of T cells in cirrhosis is more complex. While some T cell sub-populations show diminished responsiveness in chronic liver disease, others, such as memory T cells, appear less affected. This variance in responsiveness may be more pronounced with different types of vaccinations. Notably, new mRNA vaccines seem to elicit stronger T cell responses, potentially reducing the severity of infectious diseases for which they have been developed. Confirming the integrity of this immune mechanism in cirrhotic individuals could be crucial for optimizing vaccination strategies. Therefore, developing a standardized method for evaluating T cell responses is essential. Nonetheless, the impairment in innate immune responses, especially in DCs, diminishes the efficacy of adaptive immunity priming. Understanding which innate immune mechanisms are compromised and which remain functional in cirrhosis is essential for developing effective vaccines and identifying suitable adjuvants.

The disruption of the gut–liver axis could also play a significant role in the reduced vaccination response associated with cirrhosis. However, data on this topic are lacking. Gut–liver axis dysfunction and dysbiosis are associated with immune dysfunction and could also be implicated in the impairment of the immunization process. In patients with liver cirrhosis, the gut microbiota is strongly dysregulated. Alongside that, the loss of SCFA production, the alteration of gut permeability, and GALT dysregulation might contribute to the impaired vaccination response. Research in this area is currently limited, and further investigation is crucial to identify the cellular and molecular mechanisms underlying reduced immunization and to develop new strategies for enhancing vaccination efficacy in cirrhotic patients. The adjuvant effect of specific bacterial molecules, the direct stimulation of antigen-presenting cells, or an increased production of immunomodulatory agents such as SCFAs may enhance immunization. For this purpose, it would be beneficial to conduct studies that take into account various factors, such as vaccine types, cytokine profiles, and the responses of T and B cells, as well as the composition of the microbiota and their metabolites.

## Figures and Tables

**Figure 1 vaccines-12-00349-f001:**
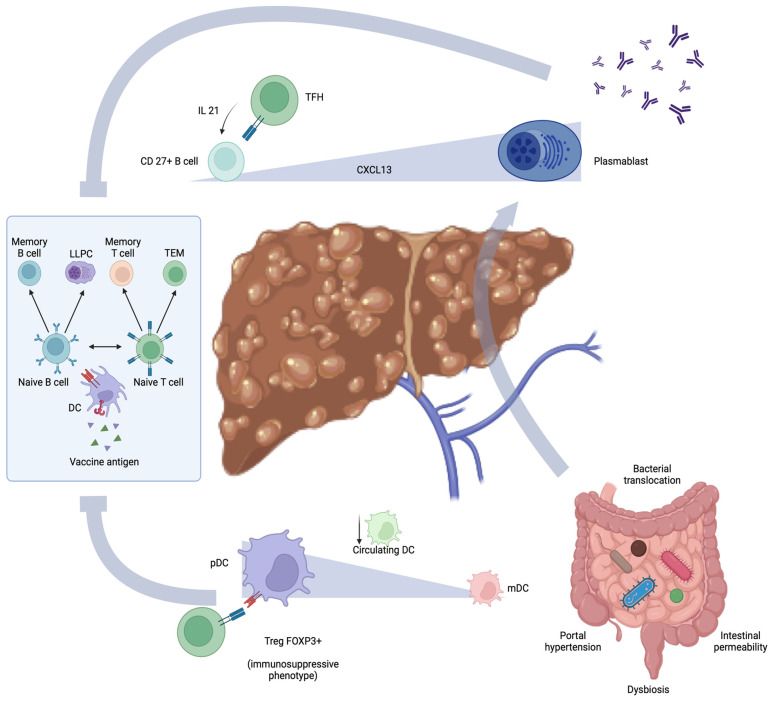
The complex interplay between vaccination, immune response, and the effects of cirrhosis on immunization. Vaccination triggers DCs, leading to the maturation of naive B and T cells. Memory B cells are primed to respond to future encounters with the antigen, while LLPCs contribute to durable humoral immunity, preventing infection. Memory T cells and TEM swiftly counteract pathogen replication at the onset of infection. In patients with liver cirrhosis, there is a remarkable shift from memory B cells towards plasmablasts, amplified via TFH activation through pro-inflammatory cytokine production. Complications, such as portal hypertension and increased intestinal permeability, together with gut dysbiosis, promote bacterial translocation, exacerbating plasma cell immunoglobulin production. This process diminishes the memory B cell reservoir, impairing the immunization response. Additionally, there is a reduction in the circulating DCs that are involved in adaptive immune response activation through vaccine antigen presentation. Furthermore, a notable transition from mDCs to pDCs, which interact with FOXP3+ T cells, fosters an immunosuppressive environment, further complicating the immune response in cirrhosis. DCs, dendritic cells. IL-21, interleukin 21. CXCL13, chemokin CXC ligand 13. LLPC, long-lived plasma cell. TEM, T effector memory cell. TFH, T follicular helper. mDC, myeloid dendritic cell. pDC, plasmacytoid dendritic cell.

**Table 1 vaccines-12-00349-t001:** Studies reporting the efficacy of anti-HBV vaccines in cirrhotic patients.

Authors	Year	Study Design	Type of Anti-HBV Vaccine	Number of Patients	Outcome
Gutierrez Domingo et al. [29]	2012	Retrospective cohort	Three-dose recombinant vaccine (Engerix B, Recombivax HB)	278 cirrhotic patients	Higher response in Child-Pugh A (54%) than in B and C (30–33%)
Roni et al. [30]	2013	Prospective cohort	Three-dose recombinant vaccine (Shanvac-B)	52 cirrhotic patients	Higher response in Child-Pugh A (88%) than B (33%); alcohol-related disease linked to poorer efficacy
Aggeletopoulou et al. [22]	2017	Review	Single or double dose of three-dose recombinant vaccine	961 cirrhotic patients included in 11 prospective and retrospective studies	The mean response rate was 38% for the standard dose and 53% for the double dose
Amjad et al. [28]	2020	Retrospective cohort	Two-dose (Heplisav-B) and three-dose (Engerix B) recombinant vaccine	166 CLD patients (34% with cirrhosis)	Higher response rate with two-dose than three-dose regimen (aOR: 2.74, 95% CI 1.31–5.71).
Kim et al. [23]	2023	Retrospective cohort	Three-dose recombinant vaccine	126 CLD patients (82% with cirrhosis)	The cirrhotic patient response rate was 51% vs. noncirrhotic, which was 72% (*p* = 0.04)

**Table 2 vaccines-12-00349-t002:** Studies reporting the efficacy of anti-HAV vaccines in cirrhotic patients.

Authors	Year	Type of Study	Type of Anti-HAV Vaccine	Number of Patients	Outcome
Keeffe et al. [32]	1998	Prospective cohort	Two-dose regimen of the Havrix vaccine	220 cirrhotic patients (n = 104 with chronic Hepatitis C; n = 46 with chronic Hepatitis B; n = 70 with other CLD)	Adequate seroconversion rate (94–98%) in patients with HBV and HCV-related nonadvanced chronic liver disease
Arguedas et al. [33]	2001	Prospective cohort	Two-dose regimen of the Havrix vaccine	84 cirrhotic patients (49 with compensated liver disease and 35 with decompensated disease)	Higher seroconversion rate (98%) in compensated cirrhosis (Child-Pugh A). As the severity of cirrhosis increased, the response rate gradually decreased.

**Table 3 vaccines-12-00349-t003:** Studies reporting the efficacy of anti-pneumococcal vaccination in cirrhotic patients.

Authors	Year	Type of Study	Type of Anti-Pneumococcal Vaccine	Number of Patients	Outcome
Pirovino et al. [34]	1984	Prospective cohort	14-valent pneumococcal polysaccharide vaccine (Pneumovax-14)	15 patients with biopsy-proven alcoholic liver cirrhosis (compared to 10 healthy volunteers and 10 patients with chronic obstructive pulmonary disease).	The response rate in patients with alcohol-related cirrhosis was similar to the other groups.
Preheim et al. [35]	1992	Preclinical study in vivo	Type 3 pneumococcal capsular polysaccharide (PCP) antigen	Rats with induced cirrhosis	Rats with cirrhosis had a substantially higher pneumococcal infection-related mortality than vaccinated healthy rats despite an adequate serological response.
McCashland et al. [36]	2000	Prospective cohort	Pneumococcal polysaccharide vaccine (PPSV23)	45 patients with end-stage liver disease (compared to 13 age-matched control subjects)	Specific anti-pneumococcal polysaccharide capsule IgA, IgM, and IgG significantly increased in both patients and healthy controls at one month without statistically significant differences. The comparative 6-month-to-baseline elevations for both IgM and IgA were significantly lower in the patient group than in the control group.

**Table 4 vaccines-12-00349-t004:** Studies reporting the efficacy of anti-influenza vaccination in cirrhotic patients.

Authors	Year	Type of Study	Type of Anti-Influenza- Vaccine	Number of Patients	Outcome
Gaeta et al. [38]	2002	Prospective cohort	2000/2001 season virosomes adjuvanted influenza vaccine (Inflexal V)	20 patients with HBV/HCV-related cirrhosis and eight age-matched controls	Seroconversion rate of 75–85% in cirrhotic patients compared to 100% in the control group.
Härmälä et al. [37]	2019	Meta-analysis comprising 12 studies (1 randomized controlled trial and 11 cohort studies; 6 with clinical outcomes, 6 with serological outcomes)	Monovalent, split virus; trivalent, split virus; trivalent, subunit.	Studies with clinical outcomes: 232 patients with CLD (148 cirrhotic patients), most with viral liver disease. Studies with serological outcomes: 8189 patients with CLD (3258 cirrhotic patients)	A noteworthy seroconversion rate (80% for the A/H1N1 strain and 87% for the B strain).

**Table 5 vaccines-12-00349-t005:** Studies reporting the efficacy of anti-COVID-19 vaccination in cirrhotic patients.

Authors	Year	Type of Study	Type of Anti-COVID-19 Vaccine	Number of Patients	Outcome
Thuluvath et al. [39]	2021	Prospective cohort	mRNA vaccines or Johnson and Johnson vaccine	233 patients (62 liver transplant recipients, 79 cirrhosis [10 decompensated], 92 CLD without cirrhosis.	Poor antibody responses in 61% of LT recipients and 24% of those with CLD. Only 40% of patients with cirrhosis showed an adequate serological response.
Bakasis et al. [40]	2021	Prospective cohort	Two doses of mRNA-based vaccinations	38 patients with cirrhosis and 49 noncirrhotic chronic liver disease compared to 40 controls.	Appropriate rates of seroconversion: 97.4% (37/38) in cirrhotics, 87.8% (43/49) in noncirrhotic liver disease, and 100% (40/40) in controls.
Iavarone et al. [41]	2023	Prospective cohort	BNT162b2 and mRNA-1273	182 cirrhotic patients (85% SARS-CoV-2-naïve) compared to 38 controls.	Anti-spike IgG serum levels were significantly lower in 182 cirrhotic patients who received two doses of mRNA vaccine than in healthy controls (1751 U/mL vs. 4523 U/mL, *p* = 0.012).
Beran et al. [42]	2023	Meta-analysis (including four studies)	BNT162b2, mRNA, mRNA-1273, JNJ-784336725, Ad.26.COV2.S, AstraZeneca, Bharat Biotech, CanSino, and Sinovac.	51834 cirrhotic patients	COVID-19-related hospitalization rate and related mortality rate were significantly lower in vaccinated cirrhotic patients compared to unvaccinated ones.

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
