# Peer review of "Vaccine Responses in Patients with Liver Cirrhosis: From the Immune System to the Gut Microbiota"

_vaccines, 2024, doi:10.3390/vaccines12040349_

Round 1

Reviewer 1 Report

Comments and Suggestions for Authors

This review addresses the clinically relevant topic of the immunological impairments contributing to the reduced vaccine response observed in individuals with cirrhosis, discussing the potential and involved mechanisms. I have only minor suggestions to further improve the clinical impact of the manuscript.

-discussing the cirrhosis-related immune dysfunctions, other important immune impairments have been reported: 1) ascitic neutrophils from cirrhotic patients with spontaneous bacterial peritonitis where bacterial-induced resistin production down-regulate the inflammatory response of macrophages and neutrophil function, as previously reported (J Leukoc Biol. 2018 Oct;104(4):833-841. doi: 10.1002/JLB.3A0218-072R); 2) the anti-inflammatory and antioxidative stress role, as well as volume-expanding properties and endothelial-stabilizing attributes of albumin are well-recognized. The presence of autoantibodies against albumin in patients with liver cirrhosis has been described, likely following oxidized forms of albumin which in turn result in neoepitopes recognized by the immune system. These alterations may have the potential to induce antialbumin immune responses and thus favor systemic inflammation (Mediators Inflamm. 2019 Feb 25;2019:7537649. doi: 10.1155/2019/7537649.); 3) the term "cirrhosis-associated immune dysfunction (CAID)" has been recently proposed and refers to refers to the dynamic spectrum of immunological perturbations that develop in patients with cirrhosis, which are intimately linked to the underlying liver disease, and negatively correlated with prognosis, 4) Patients with autoimmune disorders are characterized by an immune disregulation. To mitigate the risk of “acute-on-chronic liver failure” (ACLF) in patients with underlying autoimmune hepatitis (AIH), international practice guidelines call for vaccination against HAV and HBV to avoid an hyperacute exacerbation of AIH possibly favored by long-term immunosuppression, as described (Acute-on-chronic liver failure: A complex clinical entity in patients with autoimmune hepatitis. J Hepatol. 2021 Dec;75(6):1503-1505. doi: 10.1016/j.jhep.2021.06.035).

Author Response

Reviewer 1

This review addresses the clinically relevant topic of the immunological impairments contributing to the reduced vaccine response observed in individuals with cirrhosis, discussing the potential and involved mechanisms. I have only minor suggestions to further improve the clinical impact of the manuscript.

 We thank the Reviewer for his/her general opinion on the manuscript.

-discussing the cirrhosis-related immune dysfunctions, other important immune impairments have been reported: 1) ascitic neutrophils from cirrhotic patients with spontaneous bacterial peritonitis where bacterial-induced resistin production down-regulate the inflammatory response of macrophages and neutrophil function, as previously reported (J Leukoc Biol. 2018 Oct;104(4):833-841. doi: 10.1002/JLB.3A0218-072R);

Thanks for this comment. As suggested, we discussed the role of ascitic neutrophils in cirrhotic patients with spontaneous bacterial peritonitis in lines 490-493 of the manuscript.

2) the anti-inflammatory and antioxidative stress role, as well as volume-expanding properties and endothelial-stabilizing attributes of albumin are well-recognized. The presence of autoantibodies against albumin in patients with liver cirrhosis has been described, likely following oxidized forms of albumin which in turn result in neoepitopes recognized by the immune system. These alterations may have the potential to induce antialbumin immune responses and thus favor systemic inflammation (Mediators Inflamm. 2019 Feb 25;2019:7537649. doi: 10.1155/2019/7537649.);

Thanks for this comment. As requested, we reported the possible role of the antialbumin antibodies production in the immune dysregulation that occurs in patients with cirrhosis in lines 527-529 of the manuscript.

3) the term "cirrhosis-associated immune dysfunction (CAID)" has been recently proposed and refers to refers to the dynamic spectrum of immunological perturbations that develop in patients with cirrhosis, which are intimately linked to the underlying liver disease, and negatively correlated with prognosis,

Thanks for this comment. We have included the term CAID to refer to the immune alterations of cirrhotic patients in lines 380-383 of the manuscript.

4) Patients with autoimmune disorders are characterized by an immune disregulation. To mitigate the risk of “acute-on-chronic liver failure” (ACLF) in patients with underlying autoimmune hepatitis (AIH), international practice guidelines call for vaccination against HAV and HBV to avoid an hyperacute exacerbation of AIH possibly favored by long-term immunosuppression, as described (Acute-on-chronic liver failure: A complex clinical entity in patients with autoimmune hepatitis. J Hepatol. 2021 Dec;75(6):1503-1505. doi: 10.1016/j.jhep.2021.06.035).

Thanks for this comment. We have reported the recommendations in lines 230-232 of the manuscript, as suggested.

Reviewer 2 Report

Comments and Suggestions for Authors

I have the following suggestions

1. Please add a section on 'recommendations for vaccination of cirrhosis patients'

2. Please discuss about the response to accelerated vaccination schedule for hepatitis B, double dose vaccination, intradermal vaccination, and booster dose etc in cirrhosis

3. May discuss about the vaccine response in liver transplant recipients

4. Briefly discuss about the mortality/morbidity burden of vaccine prevnetable diseases in cirrhosis 

Comments on the Quality of English Language

Minor editing

Author Response

I have the following suggestions

  1. Please add a section on 'recommendations for vaccination of cirrhosis patients'

Thanks for this comment. As requested, we added a section on “recommendations for vaccination of cirrhosis patients” (paragraph 2.2 of the manuscript).

  1. Please discuss about the response to accelerated vaccination schedule for hepatitis B, double dose vaccination, intradermal vaccination, and booster dose etc in cirrhosis

Thanks for this comment. As requested, we discussed the response to the different protocols of HBV vaccine administration (doubled dose, intradermal administration and booster dose) in cirrhotic patients in lines 92-144 of the manuscript.

  1. May discuss about the vaccine response in liver transplant recipients

Thanks for this comment. As requested, we discussed the vaccine response in liver transplant recipients (paragraph 2.3 of the manuscript).

  1. Briefly discuss about the mortality/morbidity burden of vaccine prevnetable diseases in cirrhosis

Thanks for this comment. As suggested, we discussed about the mortality and morbidity of vaccine preventable diseases in patients with cirrhosis in lines 45-62 of the manuscript.

Round 2

Reviewer 2 Report

Comments and Suggestions for Authors

Thanks for accepting my suggestions